# Influenza A Virus Vaccination: Immunity, Protection, and Recent Advances Toward A Universal Vaccine

**DOI:** 10.3390/vaccines8030434

**Published:** 2020-08-03

**Authors:** Christopher E. Lopez, Kevin L. Legge

**Affiliations:** 1Department of Microbiology and Immunology University of Iowa, Iowa City, IA 52242, USA; christopher-lopez@uiowa.edu; 2Department of Pathology, University of Iowa, Iowa City, IA 52242, USA

**Keywords:** influenza virus, vaccine, adaptive immunity, T cells, antibody

## Abstract

Influenza virus infections represent a serious public health threat and account for significant morbidity and mortality worldwide due to seasonal epidemics and periodic pandemics. Despite being an important countermeasure to combat influenza virus and being highly efficacious when matched to circulating influenza viruses, current preventative strategies of vaccination against influenza virus often provide incomplete protection due the continuous antigenic drift/shift of circulating strains of influenza virus. Prevention and control of influenza virus infection with vaccines is dependent on the host immune response induced by vaccination and the various vaccine platforms induce different components of the local and systemic immune response. This review focuses on the immune basis of current (inactivated influenza vaccines (IIV) and live attenuated influenza vaccines (LAIV)) as well as novel vaccine platforms against influenza virus. Particular emphasis will be placed on how each platform induces cross-protection against heterologous influenza viruses, as well as how this immunity compares to and contrasts from the “gold standard” of immunity generated by natural influenza virus infection.

## 1. Introduction

Influenza virus is a segmented, negative-sense RNA virus of the family Orthomyxoviridae. Influenza viruses are divided into four primary groups: influenza A, B, C, and D. While influenza A, B, and C viruses are all capable of infecting humans, A and B are of primary concern for human disease [1] as they are primarily responsible for the seasonal epidemics and periodic pandemics that are a significant health and economic burden worldwide. Annually, during non-pandemic years, influenza A and B viruses are responsible for approximately 200,000 hospitalizations and 36,000 deaths within the United States. Furthermore, these yearly epidemics result in a potential annual economic loss of $4–12B as a result of medical costs and lost productivity [2]. This is further highlighted by the 2017–2018 influenza season, in which there were 80,000 deaths and over 650,000 hospitalizations in the US. However, these numbers only represent a fraction of the potential burden of influenza virus on human disease as only ~40% of US adults choose to be vaccinated in any given year and the vaccine had an overall effectiveness of ~36% in 2017–2018. Even so, it is estimated that the 2017–2018 influenza vaccine prevented 7.1 million influenza virus illnesses, 109,000 hospitalizations, and 8000 deaths [3]. Thus, even when not optimally matched with the circulating strains, the influenza vaccine is effective in substantially lowering the burden caused by yearly influenza virus epidemics.

One reason for the incomplete vaccine effectiveness is that influenza virus undergoes frequent mutations in its genome as a result of its low-fidelity viral RNA polymerase. This results in changes to influenza virus proteins. If these mutations confer increased viral fitness or allow escape from existing immunity, particularly neutralizing antibodies, then these mutations are selected for. Thus, there is considerable antigenic drift in the exposed surface glycoproteins hemagglutinin (HA) and neuraminidase (NA) of influenza, which are under heavy immune pressure from neutralizing antibodies. In addition to antigenic drift, periodic reassortment events as a result of coinfection of a single cell with two antigenically distinct influenza viruses can lead to selection and repackaging of individual influenza virus segments from both viruses into a novel influenza virion (termed antigenic shift). These novel influenza viruses are thought to escape the host’s existing influenza virus immunity and possess the potential to lead to influenza pandemics. Therefore, the processes of antigenic drift/shift make it challenging to vaccinate against influenza and explain the current need for yearly vaccination with proteins matched to the current circulating strains in order to achieve protection. Thus, there is no current FDA-approved, “universal” influenza virus vaccine, i.e., a vaccine that is highly protective over multiple years against the continuous antigenic drift/shift of influenza virus [4].

## 2. The Immune Response to Influenza Virus Infection

In order to develop more efficacious vaccines against influenza virus, we must first understand the aspects of the immune response induced by natural influenza virus infection. Natural influenza virus infection is considered to be the “gold standard” in generating immunity, as protection against the matched strain of influenza virus derived from natural infection can persist for life [5]. However, because of the significant morbidity and mortality caused by influenza virus, natural infection is not a suitable means to generate immunity within a population. Upon initial infection by influenza virus, the innate arm of the immune response is the first to detect and respond to influenza virus. This first line of defense contains numerous physical, chemical, and cellular mediators in the defense against infection. These include airway epithelial cells, interferons, alveolar macrophages, and NK cells that have been extensively covered and will not be detailed in this review due to constraints on space [6,7,8,9]. As the innate arm of immunity is non-specific, it is often not enough to clear influenza virus infection and therefore the adaptive arm is required and preferred to promote the long-term pathogen-specific responses that are correlative to protection and the basis for vaccination. Induction of the adaptive immune response begins when dendritic cells (DCs), such as human CD141^+^ and CD1c^+^ DC (mouse CD103^+^ and CD11b^+^ DCs), that reside in the lung acquire influenza virus antigens (Ag)/proteins and then migrate to the draining lymph nodes (LN), where they present that Ag to naïve antigen-specific B and T cells. This interaction leads to B and T cell activation.

### 2.1. B Cell Responses to Natural Influenza Infection

For B cells, their activation by antigen-bearing DCs can be assisted via innate immune cues [10,11] and can occur in a T cell-dependent or independent manner [12,13]. In addition to natural IgM antibodies, influenza-specific B cell-derived IgG and IgA generated in the lymphoid tissues of the upper and lower respiratory tract and draining lymph nodes in response to infection produce antibodies that can mediate the clearance of influenza virus and provide long-term protection against homologous influenza virus infections [14]. Antibody (Ab) responses against influenza virus have long been known to correlate with protection ever since Wilson Smith and colleagues described in 1933 that serum from influenza virus-infected ferrets as well as human serum could neutralize the virus [15]. Subsequent studies have demonstrated that both low affinity natural Ab as well as high affinity IgG and IgA antibodies generated via germinal center (GC) reactions after infection can neutralize/inhibit virus infection via various methods: these include direct binding of predominantly hemagglutinin (HA) and neuraminidase (NA) to limit the initial infection of, or spread of the infection to additional host cells, as well as the ligation of F_c_ receptors resulting in the activation of complement, Ab-dependent cell-mediated cytotoxicity (ADCC) and Ag uptake and processing [10,16,17]. Additionally, memory B cells in the steady state create a population of influenza specific cells poised for rapid and enhanced Ab responses upon future encounters with influenza virus. While these Abs are critical in preventing infections with the same strain of influenza, they also exert immune pressure on influenza virus by selecting for drift variants of influenza virus that evade recognition by the host’s current library of influenza-specific antibodies. The influenza HA head has several distinct antigenic binding sites. As mutations occur and are selected for within one site, and eventually multiple sites, this creates holes within an individual’s Ab recognition and neutralization of HA [18]. Therefore, additional adaptive immune responses, such as T cell responses, have been described to be critical for optimal cross-protection against influenza virus.

### 2.2. T Cell Responses to Natural Influenza Infection

Cellular immunity to influenza virus can be partitioned into CD4^+^ and CD8^+^ T cell responses that respectively act in coordination to boost effector responses to influenza virus and promote killing of infected cells to limit the spread of the virus. Both subsets are critical for protection against influenza virus as T cell deficiencies have been shown to delay virus elimination and impair generation of antibodies against influenza virus [19,20]. Regarding CD4^+^ T cells, these cells comprise a heterogenous group of T cells that differentiate into specific effector roles based on the infection type and cytokine milieu. Despite their ability to differentiate into a number of effector cell types, CD4^+^ T cell responses during influenza virus infection are predominated by T helper 1 (Th1), T-follicular helper cells (T_FH_), and T-regulatory cells (T_reg_) [21]. During influenza virus infection, Th1 cells predominately act indirectly, producing cytokines IFNγ, IL-2, and IL-10 that promote the effector functions of other cell types in protection against influenza virus [22]. CD4^+^ T cells also help B cells in the form of T_FH_ cells that work directly with B cells in germinal center responses to generate the high affinity influenza-specific Ab responses mentioned previously. T_regs_ have been shown to work with T_fh_ to boost the GC B cells’ response as well as limit tissue destruction and promote tissue repair [23,24,25]. Despite contributing to anti-influenza virus responses via indirect actions, there is a growing field of evidence suggesting that CD4^+^ T cells may also have a more direct role. After influenza virus infection, a population of IFNγ producing CD4^+^ T cells that express granzyme B and perforin have been described in both mice [26,27] and humans [28]. These CD4^+^ cytotoxic T lymphocytes (Th CTL) can kill target cells in an MHC-II restricted manner and have been shown to be important for protection against lethal influenza virus infection analogous to their CD8^+^ T cell counterparts. CD8^+^ T cells, as in many other viral infections, are responsible for eliminating influenza virus infected cells. Upon their arrival in the lungs following activation in the LN, influenza virus-specific CD8^+^ cytotoxic T lymphocytes (CTLs) recognize viral peptide-MHC-I complexes present on virally infected cells via their T cell receptor (TCR) and mediate killing of the infected cells using two different mechanisms: perforin-mediated granule exocytosis and binding of tumor necrosis factor (TNF) family members with their respective ligands [29]. Both mechanisms lead to target cell death via apoptotic pathways, thus mitigating propagation of infectious influenza virions and further promoting viral clearance. After clearance of a primary influenza infection, contraction of the immune response leaves a small population of influenza specific memory T cells that remain primed for rapid responses upon secondary encounters with influenza virus. As a result, this T cell memory leaves the host better protected against subsequent influenza virus infections relative to a naïve host. Despite the presence of differing T cell specificities that recognize many different viral proteins, viral clearance by T cells becomes more complex when subsequent infections occur with a heterologous influenza virus, as individual populations of influenza virus specific CD8^+^ T cells recognize a single viral epitope (typically a 9–17 amino acid peptide sequence). Therefore, these T cells can recognize fully homologous peptide epitopes that share the same sequence in the new heterologous virus. Additionally, there is evidence that a fraction of these T cells can be cross-reactive and may be able to still recognize that epitope in the heterologous virus and exert their effector function even when a limited number of amino acids in that epitope may have mutated in the heterologous strain [30,31,32]. The generation of these cross-reactive T cells during primary infection is important especially upon later infection with influenza viruses that have undergone antigenic drift and shift, leading to those viruses escaping from neutralizing Ab. Thus, CD8^+^ T cells that recognize these altered epitopes can help contribute to protection (i.e., heterosubtypic protection) against unrelated strains of influenza virus that the host has never been infected with.

### 2.3. Heterosubtypic Immunity after Natural Influenza Virus Infection

This expanded breadth of protection is critical for immunity against heterologous influenza virus strains (antigenically drifted/shifted) that the host has not previously been exposed to. Studies show that both arms of the adaptive immune response—i.e., T cells and antibodies generated after challenge with a single strain of influenza virus—can provide various levels of protection from infection with heterologous, unrelated strains of influenza virus [33,34,35]. This is a result of the generation of antibodies and T cells that are cross-reactive to related but distinct influenza virus antigens [30,36]. This pre-existing immunity against heterologous viruses has been observed several times throughout history during influenza pandemics. The most notable instance was during the 1918 influenza pandemic. Older adults born prior to 1889 fared significantly better against the pandemic influenza virus than those born after; it is theorized that this was the result of an H1 and/or N1 influenza virus that circulated within the population prior to 1889, when it was replaced by an H3 influenza virus (the “Russian” influenza pandemic of 1889–1892). Therefore, individuals born before 1889 would have had cross-reactive antibodies and immunity that protected them from the 1918 H1N1 influenza virus while younger populations would have been immunologically naïve [37,38]. Again in 2009, studies showed a role for existing Ab and T cells in protection against the pandemic virus as those born in 1900–1920 had high Ab titers against the 2009 pandemic virus [39]. Furthermore, studies showed that those individuals with existing CD4 and CD8 T cells that were cross-reactive to the new pandemic virus exhibited less symptoms and less severe disease after infection with 2009 pandemic H1N1 virus [40].

### 2.4. Influenza Virus Resident Memory

Memory cells that are generated and persist after influenza virus infection have historically been divided into two primary groups. Characterized by distinct homing mechanisms and differential functions, T-effector memory cells (T_EM_: CD62L^−^CCR7^−^) circulate between the blood and non-lymphoid tissues (NLT), where they predominate and play a similar role to their effector counterparts, producing cytokines and expressing perforin in the case of effector memory CD8^+^ T cells. T-central memory cells (T_CM_) are distinguished from T_EM_ by their differential expression of CD62L and CCR7 (CD62L^+^CCR7^+^) and circulate through secondary lymphoid tissues in addition to blood and NLT. T_CM_ are also capable of producing cytokines to promote efficient immune responses and serve as a source to replenish the effector pool of T cells upon new antigenic encounters during subsequent infections [41]. In contrast to these two subsets of memory cells, another memory subset that is critical for protection against influenza infection has also been described [42]. Adoptive transfer and parabiosis studies in mice have demonstrated a population of non-recirculating memory cells after infection, termed tissue resident-memory T cells (T_RM_) [43]. These CD4^+^ and CD8^+^ T_RMs_ have been described in a number of tissues, including the female reproductive tract, gut, liver, lung, spleen, skin, and LN [44], and are defined by their non-circulating tissue residence and expression of C-type lectin CD69 and the integrin α_E_β_7_ (CD103); however, the latter is not expressed on CD4^+^ T_RM_ and not required for CD8^+^ T_RM_ in all tissues. The role of these cells is analogous to that of Ag-experienced T cells with innate-like sensing and alarm capabilities that are able to generate rapid responses to pathogen reinfection, producing effector cytokines and recruiting cells to the site of infection more quickly than their circulating memory T cell counterparts. Like their memory T cell counterparts, they are typically long-lived; however, this is dependent on the tissue. Influenza virus induced T_RM_ are created during the initial influenza infection and can be found in both the upper and lower airways, where they play a crucial role in protection against influenza virus infection, particularly against heterologous viruses [45]. Upon secondary encounter with influenza virus, T_RM_ in both the upper and lower airways appear to mediate protection through a supportive role by rapidly producing effector cytokines such as IFNγ and promoting cell recruitment [46]. The literature on their cytolytic capabilities is mixed, as while some studies show that that they may be cytolytic, it appears that this ability is not as robust as the effector counterparts [45,46]. Globally, T_RM_ are thought to persist for long periods of time after infection. However, after influenza infection, it appears that this persistence may relate to their position in the airways and the number of prior antigen exposures, as T_RM_ in the lungs wane significantly over time, whereas T_RM_ in the upper airways persist, much like T_RM_ populations in other tissues [47]. Interestingly, a recent study has found that subsequent exposures of T_RM_ to multiple challenges affects their kinetics—that is, influenza-specific T_RM_ found in the lungs after primary influenza infection (1° memory) are less durable and wane more quickly than T_RM_ generated by multiple exposures to influenza virus (i.e., 4° memory) [48]. This finding of increased durability after re-exposure may explain why several studies have identified and characterized influenza-specific T_RM_ within human lungs as re-exposure to influenza virus multiple times during life may increase their durability in the human lung and factor into their role in protection against subsequent influenza virus infections [49]. More recently, the list of memory T cells has been refined through the discovery of another group of memory T cells termed peripheral memory T cells (T_PM_). These T cells are distinguished from other memory T cell subsets via high CX3CR1 (fractalkine receptor) expression. Relative to other memory T cell subsets, current studies suggest that T_PM_ do not express CCR7 or CD62L, are abundant in the circulation and tissues but absent in lymph nodes, poorly proliferate and produce IL-2 in response to Ag, and are efficient killers of target cells [50,51]. Relative to the other memory T cells, the role that T_PM_ play in anti-influenza virus immunity is still being explored; it is likely, though, that T_PM_ contribute to protection as they have been identified in lung tissue [50].

T cells are not the only adaptive immune cells capable of developing into tissue-resident Ag specific cells. Recent developments in the field of B cells have revealed subsets of memory B cells expressing homing receptors to mucosal tissues, including the lungs, that also express CD69, which is characteristic of T resident memory cells and do not recirculate. These B_RM_ cells are elicited following primary influenza infection and, like T_RM_, they are capable of responding rapidly to challenge infection compared to other memory B cells from lymphoid tissues [52,53]. The role of B_RM_ during influenza virus infection relative to their non-resident memory counterparts appears to be in inducing more rapid secondary antibody secreting cell (ASC) responses in the lungs. Furthermore, adoptive transfer studies suggest that they also provide more efficient protection than their memory counterparts. Overall, these findings suggest that there may be qualitative differences between B_RM_ and memory B cells and that the localization of B_RM_ may not be the only variable in their efficacy against secondary influenza virus infection [52]. Unlike T_RM,_ studies focused on B_RM_ are currently less numerous so many questions still remain about these specialized B cells, including their longevity and persistence in the tissues [54].

### 2.5. Immune Correlates of Protection Against Influenza Virus

While influenza virus invokes a myriad of responses; as described above, some of these responses are thought to better correlate with protection and are thus more desirable in limiting influenza virus infection while avoiding immune responses that may be detrimental to the host. As such, much effort has focused on determining the qualitative level of humoral and cellular immune responses that are sufficient for broad protection against influenza virus [40,55]. In regard to humoral responses, it has been noted that antibody levels, especially those targeted against HA, are known to prevent influenza virus infection and correlate with protection when levels are >1:40, as measured by HAI [55,56]. However, antibodies are also generated against other influenza virus proteins such as NA and there are data to show that these antibodies may be correlated with enhanced protection relative to anti-HA Ab immunity. Specifically, while anti-HA Ab immunity can prevent infection, anti-NA Ab immunity contributes to protection and less severe disease by restricting the release of newly synthesized influenza virions from cells [57,58,59]. The role of non-neutralizing antibodies against internal and more conserved influenza virus proteins is less clear; however, there are data to suggest that these antibodies also play an important role in anti-influenza virus immunity. For example, antibodies against the extracellular domain of the M2 protein of influenza virus (M2e) appear to aid in reducing virus titers and ameliorating disease via ADCC in mice and humans [60,61,62], while anti-NP antibodies can facilitate particle and antigen uptake and presentation, leading to reduced viral titers and morbidity [62,63,64]. For cellular responses, it has been established that the presence of both pre-existing influenza virus specific T cells and cytokine producing effector T cells and T_RM_ are important correlates of protection against influenza virus [19,40,45]. Furthermore, influenza virus-specific T cells targeting epitopes in more conserved proteins like NP, M1, PA, PB1, PB2, and M2e can provide broader coverage against a number of influenza viruses, a goal in the development of universal vaccines that will be discussed later [65,66,67]. More recently, though, research investigating these cellular responses has begun to enumerate the specific level of T cell immunity that is sufficient for broad protection against influenza virus infection. Utilizing human samples and data from the 2009 H1N1 influenza virus pandemic, multiple groups have established predictive models that identify a baseline for T cell responses that are correlated with protection against clinical influenza virus infection. Looking specifically at IFNγ producing T cells using ELIspot, the consensus suggests that ≥100 spot-forming cells per million peripheral blood mononuclear cells is correlated to a likelihood of ≥70% protection [40,68].

## 3. Vaccination

Vaccinations against influenza virus, in addition to antiviral drugs, which are not covered in this review, are currently the most important countermeasures to provide protection against influenza virus infection. However, despite being efficacious and providing robust protection when correctly matched to the present circulating strains of influenza virus, current seasonal influenza vaccines have limitations. When not matched correctly to circulating influenza virus strains, they do not provide optimal immunity. Additionally, the current platforms (i.e., IIV and LAIV) drive predominately Ab responses, with limited or no T cell responses and systemic +/− upper airway-restricted immune responses, and result in limited local immunity generated in the lungs. To begin to combat these limitations, the National Institutes for Allergy and Infectious Disease (NIAID) released a strategic plan for universal vaccination in February of 2018. This plan outlines the goal and requirements underlying a truly universal vaccine. These criteria include a vaccine that is at least 75% effective, protects against a broad range of influenza A viruses, has durable protection that lasts at least one year with a goal of longer protection, and is suitable for all age groups [69]. Before a vaccine can reach the human population, though, it must first be tested for safety and efficacy via animal models. Such research spans a number of animal models covering mice and ferrets as well as pigs and non-human primates (NHP), each with their own advantages and disadvantages. Small animal models represent an easily manipulated and reproducible animal model for influenza virus vaccine research. Mice are more easily worked with in larger quantities than other animal models and can be inbred, giving them homogenous genetics. Furthermore, they can be genetically manipulated to insert/remove genes and reagents are plentiful; thus, they serve as a valuable tool to tease out the fine details of immune responses. However, they do not transmit the virus. Ferrets are similar to mice in that they are readily available but are also susceptible to non-adapted human influenza viruses. Ferrets also exhibit clinical symptoms seen in humans such as fever, rhinitis, and sneezing and are therefore valuable in studies of transmission. However, reagents and group sizes are typically more limited and their prior exposure to influenza viruses must be determined before the start of experimental studies. Each of these small animal models has been critical in our understanding of anti-influenza immunity as well as in the development and testing of anti-viral drugs [70]. Larger animal models such as pigs serve as natural hosts for influenza viruses that can be transmitted to humans and are better representatives of the anatomy and genetics of humans relative to small animal models. However, knowledge of their prior exposure history, the availability of reagents, housing concerns, and costs serve as limitations. NHPs are the closest relative genetically to humans and thus have served as a final step in testing the efficacy and safety of many influenza virus vaccines prior to clinical trials. Despite this, large animal models are logistically difficult as a result of costs to house and maintain them in addition to ethical concerns regarding the use of primate models [71,72].

In order to understand the immune basis for a universal influenza vaccine, we will first examine current vaccines and their underlying immunological basis, covering research that spans numerous animal models and human clinical data. Specifically, we will investigate which responses these vaccines induce or do not induce and how this relates to vaccine-induced cross-protection against influenza virus. Present day influenza vaccines came to fruition after Wilson Smith and colleagues first described the isolation of influenza virus from patients and serum neutralization of the virus in ferrets [15]. Shortly after, Wilson Smith and Sir Frank Macfarlane Burnet separately discovered that the influenza virus could be grown in embryonated hens’ eggs [73,74]. These discoveries eventually led to the development of the world’s first influenza vaccine, containing a single strain of inactivated influenza A virus, by Drs. Thomas Francis, Jonas Salk, and colleagues [75]. Research investigating how to optimize the safety and efficacy of influenza virus vaccines across demographics led to the development of split and subunit influenza virus vaccines in the 1960s. Further work on influenza virus genetics by Drs. Edwin Kilbourne and James Murphy demonstrated that viral genetics could be applied to vaccine design; the ability of laboratory strains of influenza virus to replicate efficiently in a chick embryo could be combined with the desired antigenic characteristics of a newly isolated, poorly growing influenza virus through genetic reassortment [76]. A notable example of this is X-31, a reassortant of H1N1 A/Puerto Rico/8/1934 and H3N2 A/Aichi/2/1968 that possesses the HA and NA gene segments of H3N2 while its remaining segments are that of H1N1 [77]. This work was a leap in vaccine design and led to the development of the world’s first genetically engineered vaccine virus [78]. Three pandemics and nearly eighty years later, two primary types of influenza virus vaccines are presently used: inactivated influenza virus (IIV) and live-attenuated influenza virus (LAIV) vaccines.

### 3.1. IIV Vaccines

IIV is an umbrella term for vaccines in which influenza virus is grown in cell culture or embryonated chicken eggs, which are purified and subsequently chemically inactivated, generally using formaldehyde/β-propiolactone or a detergent that disrupts the virus envelope. Until recently, most IIV vaccines were developed exclusively in embryonated chicken eggs. However, some recent IIV vaccines included HA protein produced in cell lines or through recombinant technology. Presently, the IIV formulation is administered as a trivalent (TIIV) or quadrivalent (QIIV) vaccine containing antigens from two influenza A virus strains (H1N1, H3N2) and one or two influenza B virus strains respective to TIIV or QIIV. The chosen strains are meant to reflect the currently circulating H1N1, H3N2, and influenza B viruses of the Yamagata and Victoria lineages based on available data and recommendations in the US by the Center for Disease Control (CDC) Advisory Committee on Immunization Practices (ACIP). There are three types of IIV vaccines, with or without adjuvants, depending on the manufacturer: whole-virion vaccines (WIV), split-virion vaccines, and subunit vaccines. Each vaccine differs in its immunogenicity; however, for the purposes of space, we will describe the immune response to IIV in the context of split-virion influenza virus vaccines, as these are most commonly used to vaccinate humans against seasonal influenza virus [79].

#### 3.1.1. IIV Immunity

From an immunological standpoint, IIV vaccines are similar in that they are focused on generating a systemic Ab response against influenza virus after an intramuscular (i.m.) vaccination. Studies have shown that these vaccines primarily generate the majority of their protection through neutralizing and HA-targeted Ab responses, as shown through hemagglutinin inhibition (HAI), microneutralization, ELISA, and Western blot [55]. Antibodies, especially those targeted against HA, are known to prevent influenza virus infection and correlate with protection when levels are >1:40, as measured by HAI [55,56]. However, not all IIV are equivalent, as other influenza virus proteins such NA, matrix protein 1 (M1), and nucleoprotein (NP) are also present to varying degrees in IIV due to the vaccines being normalized based solely on HA content [80]. As such, the responses and protection generated by IIVs may differ and this is highlighted by renewed interest in examining influenza virus NA. Early work by Dr. Edwin Kilbourne and others on the efficacy of NA-focused influenza virus vaccines demonstrated that they were efficacious in mice and humans, protecting both children and adults [59,81,82]. These and other human studies have shown that, while anti-HA Ab immunity can prevent infection, anti-NA Ab immunity contributes to protection and lower disease severity by restricting the release of newly synthesized influenza virions from cells [57,58,59]. Furthermore, through vaccination of human subjects with either a conventional inactivated influenza virus vaccine (A/England/42/72 (H3N2) vaccine (X-37)) or an antigenically hybrid (NA-monospecific) vaccine (Heq1N2 (X-38)), Kilbourne demonstrated enhanced immunogenicity and cross-reactive Ab responses as measured by HAI and NA inhibition assays in those individuals who received the hybrid NA-specific vaccine [83]. However, the desire to prevent infection and thus symptoms, as well as the poor physical stability of NA, limited its use in IIV [84]. More recent studies suggesting that anti-NA Ab, rather than anti-HA Ab titers, may better predict protection against influenza virus beg the question, should the content of NA in IIV vaccines be revisited [85,86,87]? With regard to T cell immunity generated against IIV, de novo T cell responses are generally thought to be limited after vaccination with IIV. While limited, studies investigating cellular immune responses generated by IIV have shown that T_FH_ cells can be detected in the bloodstream after IIV vaccination and that their frequency correlates with titers of neutralizing Ab found in humans [88]. For CD8^+^ T cells, while IIV can stimulate existing memory CTL [89], its ability to activate naïve and memory CD8^+^ T cells is limited relative to natural infection or LAIV vaccination. This is the result of the limited Ag content within IIV as well as the need for that Ag to undergo an inefficient cross-presentation process after uptake by cells. This stands in contrast to the more efficient endogenous processing and presentation of Ag that occurs in virally infected cells where additionally the amount of viral protein is not as limited. Finally, by virtue of its i.m. delivery, IIV does not directly involve the nasal mucosa and lung. As such, it generates limited local tissue-resident immune responses in the airways and also does not generate sIgA [90]. Altogether, these issues impact the ability of IIV to provide heterosubtypic protection against future heterologous influenza virus infections.

#### 3.1.2. IIV Heterosubtypic Protection

Previous research has shown that both arms of the adaptive immune response are needed for optimal protection against both homologous and heterologous influenza virus strains [20,91,92]. Antibodies play a significant role in limiting infection, whereas cross-reactive T cells help to prevent severe disease. Animal models investigating the ability of IIV to protect against heterologous strains of influenza virus have shown that detectable levels of cross-reactive antibodies are induced after vaccination; however, limited cross-protection is observed [93,94]. This has been observed in humans as well, where IIV administered before the influenza virus pandemic in 2009, which contained the A/Brisbane/59/2007 H1N1 strain, did not provide protection against A (H1N1) pdm09 [39,95,96]. However, it was noted that preexisting immunity was more prevalent in adults over the age of 60 compared to children under the age of 10. The basis for this was the existence and boosting of preexisting cross-reactive antibodies in those individuals who were previously vaccinated with the 1976 swine influenza vaccine [39]; it has been noted that those who were vaccinated with the A/New Jersey/1976 swine influenza vaccine in 1976 had substantially boosted cross-reactive antibodies to the A (H1N1) pdm09 after infection or vaccination with the 2009 pandemic strain. Interestingly, these individuals also had increased frequencies of HA-stalk specific Ab [97,98]. For T cell immunity, while low-level induction of cross-protective CTL may be possible, particularly using multiple, high doses of WIV [99,100], the overall ability of IIV platform vaccines to induce heterologous protection likely relates to (1) the antigenic content of the vaccines themselves (i.e., concentration, Ag components included, or the method of virus inactivation), (2) prior influenza exposures, and (3) host genetic factors.

### 3.2. LAIV

In order to enhance cellular responses against influenza virus that more closely mimic natural infection, live attenuated influenza virus vaccines were developed. As their name suggests, LAIV vaccines refer to a whole-virion, live influenza virus vaccine in which the virus has been attenuated. These LAIVs are developed from master donor viruses (MDV) that are cold-adapted and thus temperature sensitive. The vaccine strains are cold-adapted in that they replicate most efficiently at 25 °C and are temperature sensitive in that they have a shut-off temperature of between 37 and 39 °C. This limits the virus strains in the vaccine to limited replication only in the nasal passages and does not allow spread of the vaccine to the warmer, lower lungs. The vaccine strains are made as a 6:2 reassortant, meaning that they contain the six internal gene segments of the MDV, which confers the cold-adaption and temperature sensitivity and the HA/NA of the chosen circulating wild-type (WT) influenza virus [101].

#### 3.2.1. LAIV Immune Responses

While LAIV is restricted to the upper airways, the immune responses that it drives are more diverse than IIV, invoking both humoral and cellular responses in the upper airways [101]. Within the upper airways, LAIV elicits secretory IgA (sIgA), which is the predominant Ab produced on mucosal surfaces like the nasal passages and as such is also important in protection against influenza virus infection; of note, IgA deficient mice show enhanced susceptibility to influenza A virus infection and impaired T helper cell priming [102]. Similar to inactivated vaccines, systemic IgG is also elicited by LAIV, contributing to the neutralization of influenza virus. Both IgA and IgG Ab levels are durable for at least 6–12 months after LAIV vaccination, as shown by studies in seronegative adults and children [103,104], and maybe even longer in populations that have been previously exposed to influenza virus infection, as shown by studies demonstrating protective antibodies in individuals that were protected during the 2009 H1N1 pandemic as a result of prior vaccination in 1976 with the “swine flu” vaccine and exposure to influenza virus [39,97]. In regard to cellular responses induced by LAIV, mice display a robust expansion of CD4^+^ and CD8^+^ T cells. This also leads to the production of antiviral cytokines, especially in the nasal associated lymphoid tissue (NALT) of the upper airways [105,106]. Regarding the lower airways, the literature is mixed. It has been observed in mice that vaccination with FluMist LAIV (20 uL) was able to induce protective lung T_RM_ following vaccination [107]. Meanwhile, another study vaccinated conscious mice with a lower dose of FluMist (5 uL) and showed that this did not induce a sufficient number of cross-reactive CD8^+^ T cells within the lungs to provide cross-protection [108]. Additionally, Wang et al. showed that varying the dose of LAIV (relative to focus forming units (ffu) administered in 30 uL PBS) influenced the subsequent immune response and immunohierarchy but had no impact on the generation of lung T_RM_ [109]. Thus, the development of lung resident T_RM_ in some mouse models may reflect the volume in which the vaccine is administered, allowing the deposit of the LAIV directly into the lower lungs. In the context of humans, studies comparing immune responses to LAIV in children and adults showed that only children mounted T cell responses after LAIV [110,111]. As adults have encountered influenza virus multiple times compared to children, the lack of significant de novo T cell responses in adults may be due to pre-existing immunity, such as that against the vaccine backbone of LAIV. Similar to vaccination with IIV, however, it is likely that subsequent exposures in both adults and children to influenza virus after LAIV vaccination may lead to a boosting of existing T cell responses. Finally, our current understanding of cellular responses induced in humans after LAIV are incomplete as much of what we know is based on peripheral blood samples and may not directly correlate with what happens locally in the upper airways.

#### 3.2.2. LAIV Limitations

Since it is still a live virus, the greatest limitation of LAIV is that it is not recommended for use in high risk groups, including children <2 years of age, adults >55 years of age, and the immunocompromised. Moreover, despite invoking enhanced cellular immunity against influenza virus, LAIV-induced protection is not granted in all demographics and efficacy during some years has been limited. Clinical trials in humans comparing the efficacy of LAIV and IIV suggest that, while protective in young children with limited previous exposure to influenza viruses, extensive previous exposures to influenza virus may limit the LAIV’s efficacy in adults [90,112]. However, results are mixed, with some studies suggesting that IIV is more protective and others suggesting that LAIV can be at least as protective as, if not more protective than, IIV, depending on the age group [113]. As a result of this, there have been recent years (2016–2017 and 2017–2018 influenza seasons) in which the CDC Advisory Committee on Immunization Practices (ACIP) has not recommended the use of LAIV due to variable efficacy. Furthermore, even though LAIV is capable of generating Ab responses and variable T cell responses, it may not generate the local resident memory T cells within the lungs that are thought to be critical for protection against subsequent heterologous influenza virus infections.

#### 3.2.3. LAIV Cross-Protection

Since it more closely mimics immune responses generated after natural influenza virus infection, LAIV is thought to have a greater potential to provide protection against heterologous influenza virus infections. In terms of antibodies, LAIV has been shown to generate cross-reactive immune responses to influenza virus in both animals and humans. Cross-reactive antibody responses in LAIV as well as candidate pandemic LAIV vaccines have been observed, showing protection against heterologous challenge in animals [114,115] and cross-reactive HAI in seronegative humans [116]. In terms of T cells, Zens, et al. [107] compared immune responses induced by vaccination of mice with Fluzone IIV or FluMist LAIV. In their study, they observed that FluMist LAIV was able to induce lung T_RM_ following vaccination and through FTY720 treatment of mice and that these T_RM_ were responsible for mediating protection against heterosubtypic infection. Again, though, the extent to which LAIV may induce cross-protective responses against heterologous viruses, specifically those which are T_RM_-mediated, is likely influenced by the vaccine dosage and localization as the above studies had found that cross-protective capability as well as general immunity may vary according to dosing [107,108,109]. In human studies, specifically in children, LAIV has been shown to boost the frequency of pre-existing cross-reactive T cells. In contrast, adults display little heterologous protection, as shown by only transient T cell responses after LAIV, likely due to prior exposures to influenza virus [110,111].

### 3.3. Combination Vaccination

As the currently available vaccines have their individual advantages and limitations, a rather simple solution of vaccinating with combinations of vaccinations could have merit. Arguably, this could lead to the synergizing of the immunity generated by such vaccines, promoting enhanced protection against homologous and heterologous strains of influenza virus. Rather than vaccinating with both platforms at the same time, though, studies suggest that it may be more efficacious when utilized in a prime-boost approach: priming with one influenza virus vaccine and then boosting after a certain period of time with the same or heterologous vaccine/antigen [117,118]. A heterologous immunization is usually preferred in order to circumvent neutralizing Ab. As mentioned previously, Slütter et al. found that FluMist vaccination by itself did not induce a sufficient number of cross-reactive CD8^+^ T cells to provide cross-protection [108]. However, by priming mice with FluMist and then boosting 6 months later with Listeria monocytogenes expressing the NP of IAV A/PuertoRico/8/34 (PR8) (LM-NP), they observed a significant expansion of NP-specific CD8^+^ T cells; this translated into mice being protected from lethal challenges with both homologous and heterologous influenza virus 2 months post-boost. In addition to this, the Subbarao group has also shown a similar ability to generate enhanced humoral immunity against a number of influenza virus strains using a combination of priming with LAIV and boosting with IIV in humans and NHP [119,120,121]. Similar to the observations of Slütter et al. in mice, other groups have found that a prime-boost regimen can lead to heterologous protection. In a series of trials, it was shown that subjects vaccinated with H5N1 or H7N7 pandemic IIV who had received H5N1 or H7N7 pandemic LAIV within the previous 2–4 years displayed a rapid and robust Ab response to the booster that was of high affinity and cross-reacted with antigenically distinct viruses within the same subtype [119,120]. Moreover, a similar result was observed in clinical trials where subjects were primed with DNA expressing H5 HA and boosted with H5N1 monovalent inactivated vaccine [122,123]. Such protection is suggested to be a result of elicitation of localized somatically hypermutated GC B cells responses within the mediastinal lymph nodes, as was shown in African green monkeys [121].

Among the limitations of combination prime-boost vaccination regimens is the difficulty of encouraging a patient to return for a follow up boost vaccination [124,125] as well as determining when the optimal time to boost would be. Based on the aforementioned data, boosting should likely occur after establishment of influenza virus-specific memory and contraction of the primary immune response, similar to the natural infection/reinfection cycle. However, further research is necessary to fully answer this question. The other limitations of combinatory vaccination are dependent on the vaccine/antigen used for the prime-boost. For those that utilize LAIV and IIV prime-boost, the limitations remain predominately the same as LAIV administration; as discussed above, this technique is not approved in many groups, such as children <2 years of age, adults >55 years of age, and the immunocompromised, and LAIV/IIV still runs the risk of antigenic mismatch.

### 3.4. Current Strategies for a “Universal” Vaccine

Given the limitations in the current vaccination strategies against influenza virus discussed previously, there has been renewed interest in the development of a “universal” influenza vaccine. Concentration on this effort has led to a number of advances in vaccine platforms in recent years that span multiple approaches, covering multiple targets of influenza virus. While these approaches are diverse, including both vector and gene-based strategies, for the purpose of space-saving, we will cover the three predominant, novel, protein-based platforms: HA stem targeting vaccines, M2e virus-like particles (VLP), and nanoparticle-based vaccine platforms [126,127,128].

#### 3.4.1. HA Stem Vaccines

One of the most researched platforms for a universal vaccine currently has been targeting the stem region of HA. The stem region of HA is an attractive target as it is relatively conserved compared to the more variable HA head and therefore thought to be potentially less tolerant to mutations. Moreover, influenza viruses can be assigned to two groups based on the phylogenetic similarity of the HA protein; group 1 includes H1, H2, and H5, whereas group 2 includes H3 and H7. This is important because studies have identified HA stem-specific antibodies that can react against HAs within the same group [129,130]. It should be noted that one Ab, F16, has been described that can recognize both group 1 and 2 HAs [131]. Therefore, the less variable HA stem has been thought to serve as a good target for a vaccine in order to bypass the higher mutation rate observed in the HA head domain. Furthermore, targeting vaccines for the production of antibodies against the HA stem would share an overall target protein (i.e., HA) with current vaccines which generate HA head-specific antibodies [132]. Thus, a stem-focused vaccine would generate an immune response similar to that of IIV platforms but with enhanced cross-reactivity against heterologous influenza virus strains by allowing recognition of all influenza virus strains within either group 1 or group 2 HA families. However, studies suggest that generating a vaccine that induces stem reactive antibodies may come with difficulties and drawbacks. Serum analyses of humans have shown that natural antibodies against the HA stem are subdominant and present in low levels relative to the immunodominant HA head. Additionally, some reports in humans have shown that anti-HA stem titers may correlate only with reduced viral shedding and severity of disease [133,134]. Nonetheless, to circumvent this problem of immunodominance and induce broadly cross-reactive antibodies, there have been attempts at creating headless and chimeric hemagglutinins (cHA). This involves removing the HA head or developing HA with stems derived from human viruses and globular heads from different influenza virus subtypes, respectively. In the case of headless HA, however, removing the head and transmembrane domains of HA can lead to conformational changes and subsequent loss of recognition by antibodies [135]. Fortunately, there has been success in generating stable mini-HAs that show broad reactivity across HA groups; mini-HAs have been shown to provide protection against homologous and heterologous influenza virus strains via ADCC and allow boosting of cross-protective Ab responses in NHP that have previously been exposed to influenza virus [136,137]. cHAs have also shown some success through repeated immunizations in a prime-boost regimen that allows the boosting of pre-existing H1-stalk Ab levels [138]. Studies for these universal vaccine candidates are ongoing and display various degrees of efficacy depending on the type of mini-HA or cHA generated. It should be noted, however, that there have been studies that suggest that certain vaccine-induced anti-HA stem antibodies may potentially promote enhanced influenza virus induced respiratory disease [139,140] or self-reactive antibodies (i.e., VH1-69) [141,142] due to the polyreactive profile of certain antibodies and proximity of the HA stem region to the cell membrane.

#### 3.4.2. M2e Based Vaccines

Among the other influenza virus protein targets that show promise as universal vaccine target candidates is the M2e domain of influenza virus. M2e represents an exposed region of the M2 protein, composed of 23 amino acids, and is abundantly expressed on the cell surface of influenza virus-infected cells and is found in small quantities on the membrane of the virus itself. Furthermore, M2 is highly conserved among influenza viruses, making it an ideal candidate among influenza virus surface proteins [143]. One of the drawbacks of M2e, however, is that, in its native form, it is poorly immunogenic, with only a fraction of infected people developing anti-M2e antibodies and there is little evidence to support the idea that these low titer antibodies are broadly cross-protective against heterologous influenza virus strains [144]. To enhance M2e’s immunogenicity, strategies comprising recombinant M2e-carrier constructs have been used. These include M2e displaying virus-like particles (VLPs), tetramerizing leucine zippers, toll-like receptor (TLR) ligands, and mucosal adjuvants. Since there are various M2e constructs, there are differences in the type and magnitude of immune responses generated after immunization. Neirynck et al., utilizing M2e fused to the hepatitis B virus core (HBc) protein (M2HBc), showed protection in mice against lethal influenza virus challenge after both intra-peritoneal and -nasal vaccination, and this protection was Ab-mediated and transferable in a passive immunization model [145]. Other studies have demonstrated M2e-based vaccines that can generate cellular immune responses. Eliasson et al., using an M2e-adjuvant complex linked to a fusion protein (CTA1-3M2e-DD), showed that mice were protected against lethal homologous and heterologous influenza virus challenge. This protection was described to be mediated by IL-17A production from M2e-specific memory CD4^+^ T cells [146]. Another M2e fusion vaccine utilized M2e cross-linked to influenza virus NP that effectively functioned as a polypeptide nanoparticle; this vaccine demonstrated protection mediated by humoral as well as cellular immune responses against homologous and heterologous influenza viruses [147]. Several clinical trials looking at the safety and efficacy of M2e-based vaccines in humans have also been performed or are currently ongoing. An M2e-flagellin fusion vaccine developed by VaxInnate Corp. was tested in phase I trials in healthy young volunteers aged 18–49. The study reported the vaccine to be safe at low doses (0.3 and 1.0 ug) and immunogenic in 75% of the subjects after the first dose and 96% after the second dose. However, higher doses (3 and 10 ug) of the vaccine were associated with the appearance of adverse effects in some subjects, suggesting that the toxicity of the flagellin adjuvant might be an issue at increased doses [148,149]. Another study (NCT00819013) performed by Sanofi showed that the ACAM-FLU-A vaccine, similar to the M2e-HBc developed by Neirynck et al., passed a phase I clinical trial and showed the formation of anti-M2e antibodies in the blood sera of 90% of participants and was well tolerated when given alone or with adjuvant.

#### 3.4.3. Nanoparticle-Based Vaccines

Nanoparticle-based vaccine platforms represent one of the more unique strategies toward developing a universal vaccine. They resemble their pathogen targets in size and are largely similar to other platforms in their utilization of conserved influenza virus antigens to promote heterologous protection. Many of them even utilize the aforementioned HA-stem and M2e platforms within their vaccine [150,151]. However, as their name suggests, these antigens are loaded into biocompatible and biodegradable inorganic or polymeric nanoparticles that possess unique physicochemical properties; based on these properties, nanoparticles can have their size, solubility, surface chemistry, and hydrophobicity tuned to tailored biological properties [152]. One advantage of nanoparticles is that they have the potential to make vaccine design faster and more efficient as they possess a “plug and play” ability not reliant on traditional methods of vaccine production and so antigens and adjuvants can easily and quickly be swapped out. Hence, many groups have looked to formulate antigens with nanoparticles. Depending on the physicochemical properties of nanoparticles and the method by which Ag is loaded (encapsulation or conjugation), research has shown that this may act to protect the antigen payload to various degrees, improving Ag delivery to APCs, enhancing the immunogenicity of the vaccine, and creating a local Ag depot effect [152,153,154], thus mimicking the Ag persistence observed during influenza virus infection [155,156]. The immune responses generated by nanoparticle vaccines against influenza virus are heterogeneous due to the aforementioned tailoring of the physicochemical properties of the nanoparticles. Moreover, based on these properties, nanoparticle vaccine platforms can be divided into three primary groups: biologic nanoparticle vaccines (i.e., structural and/or immunologic backbones such as ferritin, norovirus P particle, and chitosan) and polymeric nanoparticles subdivided into hydrophobic and hydrophilic nanoparticles. Natural proteins like ferritin have been a source of interest as they are natural, biocompatible compounds and they self-assemble into a protein cage well suited for antigen presentation and immune stimulation. Yassine et al. [150] stabilized H1 HA stems (HA-SS) complexed on ferritin nanoparticles and vaccinated mice and ferrets in combination with a Ribi adjuvant. Performing immunization studies, they found that both mice and ferrets produced broad serum Ab responses to influenza virus and protection against heterologous H5N1 challenge, similar to studies of other HA-stem vaccines. Other groups using synthetic or natural polymeric nanoparticles encapsulating various influenza virus antigens in pigs and mice have also shown protection. Hydrophilic synthetic (d,l-lactic-co-glycolic acid) (PLGA) nanoparticles targeting influenza virus M2e have been observed to induce both Ab and cellular-mediated protection in pigs against H1N1 [157] while mice immunized with natural polymer nanoparticles based on polysaccharides (i.e., chitosan) display protective titers of local IgG and sIgA [158]. In the above study by Hiremath et al., utilizing PLGA nanoparticles encapsulating norovirus P particles containing M2e derived from a swine H1N1 influenza virus, the PLGA promoted cross-presentation by DC and resulted in robust IFNγ production by both CD4^+^ and CD8^+^ T cells [157]. Hydrophobic nanoparticles also show similar properties, albeit with the benefit of an antigen depot effect. Research by our own group as well as others has shown that, through the addition of CpG into polyanhydride nanoparticles encapsulating HA and NP proteins from an H1N1 influenza virus, cross-presentation of viral protein payloads can be induced in DCs, leading to robust CD8^+^ T cell responses. Furthermore, this vaccine was shown to promote the formation of lung and nasal resident CD4^+^ and CD8^+^ T_RM_ in addition to local and systemic humoral immunity in mice, leading to protection from homologous and heterologous influenza viruses [159]. Similar polyanhydride vaccines utilizing natural influenza antigens as well as recombinant antigens such as the H5 HA trimer (H5_3_) have also been shown to generate protective cellular responses in other animal models as well as in pandemic models of influenza virus infection [160,161]. Beyond pre-clinical animal studies, there are a number of nanoparticle influenza vaccines that have made their way to human trials and have shown promise thus far. NIAID’s Vaccine Research Center (VRC) has been testing a number of potential universal influenza candidates in phase I clinical trials. One of these is a vaccine composed of the HA stem domain of an H1N1 influenza virus that has been fused to ferritin from Helicobacter pylori (H. pylori) that naturally self assembles into a protein cage nanoparticle (H1ssF; VRC 321). While studies are not set to conclude until the end of 2020, these particles were previously shown to be immunogenic and elicit neutralizing antibodies against influenza virus in mice and ferrets [150,162]. Furthermore, a similar vaccine utilizing an H2 HA influenza head (VRC 316) was shown to be safe and well tolerated in the preliminary results of a phase I clinical trial (NCT03186781). Additionally, Novavax’s NanoFlu^TM^, a recombinant HA nanoparticle vaccine formulated with a proprietary saponin-based adjuvant (Matrix-M^TM^) [163], recently met its primary endpoints in a phase 3 clinical trial measuring the immunogenicity of the vaccine relative to QIV in adults aged 65 and older. This study is presently ongoing and is set to be completed in November of 2020.

As is the case with the immune responses that they induce, the physicochemical properties are also responsible for the limitations of certain nanoparticle vaccines. For example, while polyesters such as PLGA may be favorable due to their biocompatibility, they are hydrophilic and thus their mechanism of hydrolysis via bulk erosion allows water penetration throughout the material before the release of the particle payload. As a result, this can affect the stability and release of the encapsulated proteins through degradation and denaturing of the nanoparticles themselves, diminishing the activity and epitope availability of the protein antigens [164]. In contrast, hydrophobic nanoparticles composed of acid-catalyzed polymers such as poly (ortho esters) and polyanhydrides degrade via surface erosion, increasing control over the release kinetics of the encapsulated antigens. Again, though, the physicochemical properties determine the biological interactions and, although possessing more controlled release kinetics, poly(ortho esters) are more biologically inert than other classes of polymers and therefore are not potent stimulators of the immune response and as such often require immunogenic adjuvants [165].

As a result of being able to induce influenza-specific CD8^+^ T cells, nanoparticle-based influenza vaccines have shown significant heterosubtypic cross-protection against heterologous viruses [150,157,159]. The above studies all observed protection against a number of heterologous influenza viruses, including H1N1, H3N2, and H5N1 viruses. This is likely a result of choices regarding nanoparticle antigen composition and adjuvants, allowing some of these vaccines to promote cross-reactive CD8^+^ T cells and in some cases the generation of T_RM._ Data on the cross-protective ability of nanoparticle influenza virus vaccine platforms in humans are currently limited due to the fact that the vaccines that have reached human testing are still in early clinical trials assessing safety and initial efficacy.

## 4. Conclusions

Influenza virus vaccination is an ever growing, ever changing field as we learn more about the viruses themselves, immune correlates of protection, and the vaccines that we presently use against influenza virus. Observations from studies of natural influenza virus infection have shown that, in order to succeed in developing a broadly protective “universal” vaccine against influenza virus, protection derived from both arms of the immune response will likely be crucial. Induction of local humoral responses generating broadly neutralizing antibodies and high affinity memory B cells are important for protection against heterologous influenza viruses and in supporting other methods of viral control such as ADCC. In concert with the generation of memory T cells, especially T_RM_ capable of rapidly responding and recognizing a broad range of influenza virus epitopes, a “universal” vaccine may then protect against influenza virus infection from multiple angles, limiting the avenues by which antigenic drift/shift can evade host immunity.

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
