# Peer review of "Influenza A Virus Vaccination: Immunity, Protection, and Recent Advances Toward A Universal Vaccine"

_vaccines, 2020, doi:10.3390/vaccines8030434_

Round 1

Reviewer 1 Report

I have now read the review manuscript (vaccines-848133) entiteled: “Influenza virus vaccination: Immunity, protection and recent advances toward a universal vaccine” by Lopez CE and Legge KL.  The authors have presented several interesting views on different options for obtaining broader or cross-reactive influenza immunity through vaccination. The Universal vaccination issue is a highly desirable result of vaccination against pathogens, such as respiratory tract viral infections, and the authors have done a good job in trying to cover this topic in a small space of text. However, I miss a discussion or paragraph on what level of immunological response is required that should protect from disease or severe illness? (Infection genetics in vulnerable patient cohorts, i.e Kenney AD et al. Annu. Rev. Genet. 2017, 51, 241). Is it enough with detectable immunity, or will there be a need for a certain level of specific (or unspecific) anti-influenza response or quality to obtain protective immune response?

Specific questions:

Q1. Combination influenza vaccines, where a combined immunization is questioned based on murine models. A heterologous prime-booster procedure is suggested, but there is mainly a discussion on the problems with vaccine recipients being available at more than one time-point. Instead, the real challenge is to be able to describe at which time-interval the prime- and boost immunizations should be given? How should this time schedule be determined?

Q2. What are the advantages/disadvantages with data obtained in the mentioned animal models (mice, pigs, non-human primates) and their relation to clinical data?

Q3. The levels of protective immunity is normally based on the HAI titer in serum. What levels of qualitative T memory cells would be sufficient to provide protection?

Q4. The T-cell memory response cell phenotypes are discussed, but towards which viral proteins/epitopes should they be directed against to provide sufficiently broad immunity (to fit the universal vaccine design?)

Q5. Are there any genetically pre-determined weak or high-risk populations, for highly pathogenic influenza strains? Can these groups be identified by monitoring the T-cell memory populations in various tissues?

Q6. In the paragraph concerning Universal vaccines and Nano-particle vaccines the authors point out several combination vaccines, such as influenza M2e or HA-stem antigens, often coupled to H.pylori or Norovirus-proteins (References 140, 133 and 145). Thus, for efficient influenza-specific boosting require immunity towards these other pathogens for successful generation of broad influenzavirus immunity?

Reference 115? Missing information?

Reviewer 2 Report

In the current review, the authors summarize immunity to influenza A virus infections in general as well as current vaccine strategies (IIV/LAIV) and selected novel vaccine platforms. Those novel strategies include HA stem targeting, M2e VLPs, and nanoparticle-based platforms.

Overall, the review is well written and easy to read. However, the focus of the review is a bit odd. The first sections (Introduction/Immune responses to IAV/Recent vaccination strategies) describe the respective topics in a broad and fundamental manner, while lacking depth at some points. A good example for this is the line 58-73, where, for example, pulmonary DC subsets could be defined easily. This lack of depth is of course the consequence of the very broad focus of the review but it might provide a good overview about the topic for scientists new to the field. However, in that case the last section (current strategies for a universal vaccine) should also provide a broad overview instead of focusing on three selected approaches. Particularly, as the authors mention the important role of cross-reactive T cells and pulmonary TRM, gene- and vector-based approaches classically used to induce T cell responses should be covered in this review as well. The authors also mention the current SARS-CoV-2 pandemic in the conclusion, where gene- and vector-based vaccines are among the leading vaccine candidates. Thus, including those vaccination platforms in the current review is mandatory in order to meet the broad focus of the first manuscript.

Minor points:

  • Title: As the review mainly focuses on IAV, it would make sense to edit the title to “Influenza A Virus vaccination: …”
  • Line 36-40 and line 224: A vaccine efficacy of 40-60% in a seasonal setting with little mismatch cannot be considered as “effective” compared to other licensed vaccines (measles or rubella for example with >90%). The low vaccine efficacy especially in the elderly is a major problem of IAV vaccination.
  • Line 78: “… antibodies that can act in numerous ways…”: this statement provides virtually no information. A few lines later, those Ab activities are then explained. The whole paragraph could be more informative and concise. For example by deleting “historical” facts and giving more information about IgG subtypes, secondary Ab effector functions etc.
  • Line 305: Limited local tissue-resident immune responses but also no sIgA! Please add!
  • Line 355-356: Please add “in mice” somewhere
  • Line 427: There were different IAV pandemics in the past, please specify that it was the pdm09!

Round 2

Reviewer 2 Report

Most of my feedback was considered. Interpretation of the vaccine efficacy (good/bad) is of course a matter of debate.

I still think that the manuscript would greatly benefit from mentioning conventional T-cell inducing vaccine platforms. Particularly due to the broad and general focus of the review in the first sections.

However, the manuscript is suitable for publication in its recent form.